# Enhancement of Mechanical Properties of Pure Aluminium through Contactless Melt Sonicating Treatment

**DOI:** 10.3390/ma14164479

**Published:** 2021-08-10

**Authors:** Agnieszka Dybalska, Adrian Caden, William D. Griffiths, Zakareya Nashwan, Valdis Bojarevics, Georgi Djambazov, Catherine E. H. Tonry, Koulis A. Pericleous

**Affiliations:** 1School of Metallurgy and Materials, University of Birmingham, Birmingham B15 2TT, UK; J.A.Caden@bham.ac.uk (A.C.); W.D.Griffiths@bham.ac.uk (W.D.G.); ZSN584@student.bham.ac.uk (Z.N.); 2Centre for Numerical Modelling and Process Analysis, University of Greenwich, London SE10 9LS, UK; v.bojarevics@gre.ac.uk (V.B.); g.djambazov@gre.ac.uk (G.D.); C.Tonry@gre.ac.uk (C.E.H.T.); k.pericleous@gre.ac.uk (K.A.P.)

**Keywords:** ultrasonic treatment, contactless sonotrode, strength, elongation, degassing, cavitation, Weibull modulus

## Abstract

A new contactless ultrasonic sonotrode method was previously designed to provide cavitation conditions inside liquid metal. The oscillation of entrapped gas bubbles followed by their final collapse causes extreme pressure changes leading to de-agglomeration and the dispersion of oxide films. The forced wetting of particle surfaces and degassing are other mechanisms that are considered to be involved. Previous publications showed a significant decrease in grain size using this technique. In this paper, the authors extend this research to strength measurements and demonstrate an improvement in cast quality. Degassing effects are also interpreted to illustrate the main mechanisms involved in alloy strengthening. The mean values and Weibull analysis are presented where appropriate to complete the data. The test results on cast Al demonstrated a maximum of 48% grain refinement, a 28% increase in elongation compared to 16% for untreated material and up to 17% increase in ultimate tensile strength (UTS). Under conditions promoting degassing, the hydrogen content was reduced by 0.1 cm^3^/100 g.

## 1. Introduction

The metal casting industry and academic communities are extremely interested in improving melt quality. Microstructural refinement can be achieved, for example, by gating system optimization or melt inoculation [1,2,3]. Another promising route is the ultrasonic treatment (UST) of liquid metal. This method provides alloys with degassing, filtration and grain refinement [4,5,6,7,8]. Instead of the traditional immersed sonotrode, the contactless electromagnetic probe was recently developed to avoid melt contamination by probe damage due to corrosion in more reactive melts and to treat larger volumes of metal [9,10,11].

During processing, pressure vibrations induced in the melt by an external induction coil lead to acoustic resonance in the liquid alloy. This leads in turn to the oscillation of entrapped gas bubbles followed by their eventual collapse, the phenomenon of cavitation. Cavitation requires an ultrasonic pressure intensity larger than the cavitation threshold [5,6], and its presence is desirable in the melt as it leads to beneficial changes in the finished product.

The first observed benefit is that of degassing [5,6,12,13]. Usually in molten metal, some dissolved gases are present, for example, we can expect the presence of hydrogen in aluminium due to its high solubility [14]. Gas bubbles are formed on nucleation sites (for example oxide particles) and grow by the diffusion of dissolved gas from the melt into the bubble. In a solidified casting, bubbles become pore defects reducing the strength of the metal. 

In the presence of ultrasound waves, bubbles oscillate in size, in response to the changing acoustic pressure in the liquid. The bubbles’ behaviour in the sound field is governed by the Bjerknes forces [15]. Those forces move the bubbles up or down the pressure gradient created by the sound waves, or cause them to accumulate at pressure nodes [15]. As a result, the bubbles that have not collapsed can coalesce and, due to buoyancy, float to the surface where the hydrogen is released into the atmosphere.

The second benefit is that of structure refinement. In the right conditions the bubbles will not only oscillate but finally collapse. This event leads to extreme local pressure changes due to the shock waves and high-speed jets produced [5,6,16]. Such pressure changes are expected to be large enough to cause mechanical stress in intermetallic crystals or oxides, causing fragmentation by cracks due to brittle fracture [17]. After de-agglomeration, dispersion occurs. In addition to these local high-speed jets, the whole volume of metal is also stirred. 

The source of stirring is the time-averaged Lorentz force induced in the melt by the electromagnetic field generated by the contactless sonotrode [10]. The flow will disperse the de-agglomerated particles, which can then serve as nucleation sites facilitating heterogeneous crystallization [7,18,19,20,21]. This process is commonly known as the “activation” of impurities where the combined effect of dispersion and forced wetting, due to pressure differences in the liquid [5,6,7]. Finally, emerging crystals can also be broken by the shock waves reducing the grain sizes [5,6,7,22]. Under sonication, the growth of dendrites will be restricted, assisting equiaxed growth. As we treat the metal prior to solidification in the experiments shown here, this effect is less significant.

The improvement of the metal microstructure caused by contactless ultrasonication has been systematically reported [10,11,23,24,25,26]. The Hall–Petch equation [27] predicts that, as the grain size decreases, the Yield Strength increases. The strength is reduced by the porosity [14]. As a consequence, the reduction of the grain size and gas content due to ultrasound treatment result in metal strengthening. Previously observed grain size reduction occurring in pure aluminium and alloys inoculated by a grain refiner (introduced in quantities below those commercially used) assist in heterogeneous cavitation [10,11,23,24,25,26].

To investigate the metal quality, the Yield Strength (YS) or Ultimate Tensile Strength (UTS) and percentage elongation (L%) are commonly found by testing metal samples. Statistically, the fracture is described by the Weibull distribution based on the weakest link theory [28,29]. The cumulative probability function of the two-parameter Weibull distribution is, therefore, expressed as follows (Equation (1)):P = 1 − exp [−(σ σ_0_^−1^)^m^],(1)
where P is the probability of failure at a given property (stress, strain, fatigue life, etc.), σ, or lower. The parameter σ_0_ is a distribution scale parameter, and m is the shape parameter. The shape parameter is also known as the Weibull modulus, a parameter that reflects how much the data are scattered, with a higher Weibull modulus meaning a lower probability of fracture under stress. 

This approach has been adapted for metallurgy and has been widely used [29,30,31]. The Weibull modulus, established from the tensile strength for gravity-filled castings, is generally thought to range between 10 and 30. For aerospace castings, it is expected to be between 50 and 100 [32]. For example, it is about 73.8 for a ductile steel 1018, 91.4 for an aluminium alloy AL 7075-T651 and can be as high as 124.0 for Al 6061-T651 alloy [33].

This paper shows recent tensile test results (UTS and elongation) of ultrasonicated metal, demonstrating improvements in the cast metal quality. Degassing effects and microstructure refinement are interpreted to analyse the main mechanisms involved in alloy strengthening.

## 2. Materials and Methods

Casting experiments were conducted in a cylindrical clay-graphite crucible with an external diameter of 170 mm, an internal diameter of 135 mm and a height of 320 mm. For each experiment, the crucible was filled with about 8.5 kg of metal—a commercial purity aluminium (CP-Al) with the addition of 0.15 wt.% Al-5Ti-1B grain refiner to increase the potential number of nucleation sites for cavitation. The amount of grain refiner is small enough to make changes in the microstructure easily observable. The prototype “top-coil” (a “first generation” contactless sonotrode) [9,10,11] was used for the sonication of the liquid metal. During processing, the ambient ultrasonic noise emitted around the crucible was recorded by an Ultramic^®^ 200K (Dodotronic, Castel Gandolfo, Italy) digital ultrasonic microphone. 

The recorded sound was observed in the form of an FFT (Fast Fourier Transform) sound spectrum extracted in real-time during experiments using MATLAB^®^ Online (MathWorks, Natick, MA, USA) software. The broadband noise emitted by collapsing bubbles [23,34] acted as an indicator of the presence of cavitation. Broadband noise was seen as light-coloured vertical lines on spectrograms recorded under varying conditions. The lines were normal to the continuous horizontal lines denoting the top-coil frequency signal, observed at around 20 kHz, and the induction furnace signal, observed at around 5 kHz. Cavitation was seen to be intermittent, and the number and density of vertical lines was considered to be a good indication of cavitation activity [23,35]. 

Where broadband noise was observed, the process conditions (coil frequency and melt temperature) responsible for the noise were maintained and at least 1–2 min of processing was recorded and presented on the spectrograms. In some cases, the local conditions in the setup resulted in near-resonant conditions in the crucible, where bubbles could oscillate continuously at their own resonant frequency but did not implode. By ‘local conditions’ we mean the parameters such as the melt volume or the off-axis position of the crucible relative to the top-coil head and others. When that happened, only degassing was observed. 

The results of that degassing were compared with processing followed by grain size change. In each case, after 4–5 min of processing, samples were taken using the KBI ring test [36]. For this test, the liquid metal was poured into a steel ring with an outside diameter of 75 mm, inside diameter 50 mm and height of 25 mm placed on an insulating silica brick. Typically, the mass of aluminium samples is 70–100 g. To characterize the grain size, the base of the cylindrical samples were removed to about 3 mm above the base and ground, polished and etched with either Poultons’ or Kellers’ solution. 

The average grain size was then determined by the mean linear intercept method after taking photographs using a Zeiss Axioskop 2 (Zeiss, Oberkochen, Germany) microscope equipped with an AxioCam HRc camera (Zeiss, Oberkochen, Germany). To show the samples with a larger magnification, the standard microstructure photographs were taken by a Fuji camera (Fujifilm, Minato CITY, Tokyo, Japan) with optical zoom.

Following the KBI ring test, a second sample was poured into a Severn Science Gas Analyser (MechaTech Systems Ltd., Thornbury, Bristol, UK) to determine the hydrogen content. The remainder of the processed metal was poured into a sand mould (See Figure 1) in order to obtain tensile test-bars, which were later machined to the size required for the tensile strength tests.

In each experiment, one mould was cast, and 10 tensile specimens were produced. Producing more than 10 test-bars in one experiment is difficult due to the low initial temperature used. It is easier to release dissolved hydrogen from the melt at lower temperatures; hence, the cavitation intensity decreased with the temperature increase and almost disappeared at 720 °C. Thus, the processing happened at 710 °C. That low temperature shortened the time frame for casting into the mould. Due to fluidity loss with time, filling two moulds was difficult.

The cast metal was filtered using a 20 ppi filter to avoid the presence of excessive oxide films. The process was repeated without ultrasound processing, to obtain reference test-bars. In that case, instead of the ultrasonic processing, the liquid metal was left inside the induction furnace at the same temperature as before for the time usually required for the contactless sonotrode treatment. Both the reference and processed samples were tested with an applied strain rate of 1 mm/min by the Zwick/Roell Z030 Universal Mechanical Tester (Zwick Roell Group, Ulm, Germany), equipped with a micro-extensometer. 

The data obtained were analysed with the help of scripts in MATLAB and R-Studio software. To establish the Weibull modulus by the linear regression method [37,38,39,40], unbiased estimators were used following the approach described in [37] depending on the sample size (n—number of measurements). The general form of the estimator used in that procedure was (Equation (2)):P = (i − a)(n + b)^−1^,(2)
where i is the rank of the data point in the sample in ascending order, n represents the sample size, and a and b are numbers specific for the sample size found by the computer simulation [37].

For a sample size n = 20, that estimator is in the form (Equation (3)):P = (i − 0.417)(n + 0.030)^−1^,(3)
while for n = 10, it is (Equation (4)):P = (i − 0.348)(n + 0.190)^−1^.(4)

Where other estimators were used, the estimator form is given in the text. The coefficient of determination (R^2^) for each regression line is also presented.

The error was estimated by the simulation in MATLAB. The Weibull modulus M′ was found by linear regression after sampling 10 or 20 unique and random values from the distribution with the chosen shape parameter m = M. This procedure was repeated 50,000 times, and the mean value and the standard deviation of the mean were extracted. The ratio of M′ (found from 10 or 20 values) and Weibull modulus M set as a “true” value in the program, was treated as the bias of the method (compare: [38]).

## 3. Results

### 3.1. Grain Refinement after Sonication

Experiments denoted 1 and 2 are the principle results in this paper. The effects of the first treatment (Experiment 1) are presented below in Figure 2. The grain size of a processed metal (b,d) is compared with the initial grain size observed in the untreated metal (a,c). As can be seen in the image without microscopic magnification (c,d) the grain size decreased strongly. From the photographs taken in the microscope, the decrease was of about 48% from the initial size. The exact grain sizes are given in Figure 2.

In the next experiment (denoted experiment 2), another set of test-bars was cast, and the grain sizes of the metal before and after treatment are presented in Figure 3. The initial grain size was similar to the previously prepared reference set (Figure 2c). In this case, the grain size decreased by about 28% from that value.

The decrease in grain size is the effect of metal ultrasonication, which appears to be less effective in this case. The contactless sonotrode frequency was set in near-resonance conditions that were unique for each experiment. The spectrograms of the sound recorded near the crucible indicated the extent of cavitation during both experiments and are shown in Figure 4. 

The vertical light lines, perpendicular to the sonotrode signal seen as a horizontal line at 20 kHz, can be treated as an indication of cavitating bubbles [23,34]. In both cases, almost all the treatment time was recorded. In the first experiment, we see that the cavitation intensity appeared to be higher at the end of the process. This could be an explanation as to why the grain size in this experiment decreased by 48%, while in the second case, the decrease was only 28%. It is not unusual for the effects to be different in each experiment. 

The cavitation intensity depends on the local conditions, including the liquid volume, temperature, crucible position, crucible wall material etc., as this is the frequency that causes the resonance [34]. For industrial use, keeping those conditions constant would be possible in a specially designated set-up using a feedback mechanism; however, in this experimental set-up, it is difficult. Each experiment is, therefore, unique, and the near-resonant frequency is found in real-time by the operator observing the FFT of the recorded noise. Thus, the cavitation intensity differs in individual treatments, and the observed grain size reflects that intensity.

### 3.2. Gas Content Decrease Due to Sonication

In both the experiments presented above, the degassing potency of the contactless sonotrode was determined. The gas content before and after processing is given in Table 1 and compared with results of similar experiments in which only the degassing level was determined. The two experiments described before, in which test-bars were produced (Figure 2, Figure 3 and Figure 4), are labelled as Ex1 (experiment 1) and Ex2 (experiment 2) in this paper.

Both experiments (Ex1 and Ex2) were accompanied by a decrease in the gas content of about 0.05 cm^3^/100 g. As can be seen in the example of experiment Ex4, the degassing can be much stronger. The maximum change (Ex4) of the gas content was about two-times greater than in experiments Ex 1–3. Slightly less of a degassing effect in Ex2 than in Ex1 can be associated with the smaller cavitation intensity, manifested by a smaller grain refinement effect (compare with Figure 2, Figure 3 and Figure 4).

Previous examples of reported gas content decreased due to traditional ultrasound processing from 0.35 to 0.17 cm^3^/100 g [13]—a reduction of 50%. In experiment 4, the gas content decreased by 3.75 times. Unfortunately, the alloy used in [13] and the processing conditions were different; however, the efficiency of the contactless sonotrode in degassing seems to be at least comparable.

To summarise, the results presented show that contactless sonication provides: (I) grain refinement, and (II) degassing. Then, the open question is, how does the ultrasound treatment improve the strength of processed metals? As mentioned in the introduction, both effects—the grain size decrease and the gas content reduction should improve the cast metal quality. To determine how the described changes influenced the strength of the alloy in Ex1 and Ex2, in which elongation and UTS were measured, in Table 2, the mean values of elongation before and after treatment are given.

### 3.3. Elongation after Sonication

The mean elongation after treatment was 52–73% greater than that observed for the same alloy without sonication. These results are consistent with previous findings—in experiment 1, we observed a greater grain size decrease and slightly more degassing than in experiment 2. This was followed by better metal quality. In comparison with the degassed only samples (Ex5), an assumption can be made that the degassing at this level played a negligible effect, and thus the main mechanism in the improvement is associated with the grain size decrease. Aluminium belongs to the class of ductile materials, and an improved elongation is important for future applications of the metal.

As mentioned before, each unique experiment produced a casting of 10 test-bars. For the reference set, the temperature history of the melt was repeated, and we were able to produce equivalent tensile specimens. Thus, the reference values of both (UTS and elongation) tests were established from 20 results. As expected, the observed error was smaller in that case than for experiments Ex1 and Ex2, where only 10 specimens were tested, as can be seen in Figure 5.

Even taking into account the error shown in Figure 5, both measured values for metal samples cast after ultrasonic treatment were much higher than those observed for the reference samples. The two left arrows in Figure 5 show the minimal observed differences. The next two arrows represent differences between the mean values. The mean elongation increased maximally by about 11.5%, which makes it 1.72-times greater than that observed for the reference specimens.

### 3.4. Ultimate Tensile Strength (UTS) after Sonication

In Table 3, the mean UTS before and after treatment is presented. 

The mean UTS before and after treatment was greater than the reference set increasing from about 7% to 17%. The metal cast in the experiment resisted, without permanent damage, a pressure of about 12 MPa greater than the reference samples. To help understand these results, the Weibull distribution was fitted into the UTS data. Weibull plots of the ultimate tensile strength (UTS) data of the castings are presented in Figure 6.

From the regression line equations, the Weibull modulus (m) of the reference set equalled m_0_ = 30.5. After treatment, this modulus increased to m_1_ = 67.5 in Ex1 and m_2_ = 56.0 in Ex2. The data were fitted to Weibull distributions of m_1_ and m_2_ (and calculated scale parameters), and the curves of the Weibull probability plots are presented in Figure 7.

The shape of the distribution curve attributed to data from Experiment 1 shows that the expected scatter of the results was much smaller than that for the reference set. This is indicated by the narrower Weibull curve governed by the shape parameter (m, also known as the Weibull modulus). The results of Experiment 2 are better than those observed without treatment, and the Weibull curve is also “narrow”. Changing the estimators does not change the Weibull distribution significantly (see Section 4.3).

### 3.5. The Stress–Strain Curves

Figure 8 and Figure 9 present the stress–strain curves for all specimens, including both reference and treated (Ex1 and Ex2) metal.

In Figure 8, the failure region for the reference set was observed at much lower values of strain, and the maximum stress was higher than for untreated metal. Toughness, the ability of a material to absorb energy and plastically deform without fracturing, is defined by the area under the stress–strain curve [41]. The area under any of the experimental lines was larger than under the reference lines. The toughness of the material after ultrasound treatment in Ex1 was much higher than for the non-processed samples, which is a good prediction for future engineering applications.

In the case of Ex2, the failure in most cases occurred much later than for the reference test bars. Two specimens broke out earlier. This can be caused by other effects, such as porosity or entrained oxide films. The overall effects of ultrasonic processing were positive and prove the efficacy of the contactless technique. The toughness of the material was also significantly improved. For most cases, the area under the curves was much larger than that recorded for the untreated metal. The variability between both experiments can be attributed to the slightly different processing conditions, which were manually controlled. It is necessary to develop a feedback mechanism continuously adjusting the coil frequency for resonance and an accurate pressure monitoring system to control conditions more precisely in an industrial situation.

## 4. Discussion

### 4.1. Dataset Validity

Before comparing the results, Student’s t-test was performed to decide if separately cast test-bars could be treated as coming from the same distribution. The first check confirmed that all samples produced as reference test-bars (obtained from two separate castings with repeated conditions) belonged to the same Weibull distribution and could be presented as one dataset (red triangles in Figure 6). The same test made for both experiments excluded the possibility that the data from Ex1 and Ex2 belonged to the same Weibull distribution. The results of that test (for Ex1 and Ex2) are shown in Figure 10.

The minimum p-value to accept the hypothesis that both experiments came from the same distribution was *p* = 0.05 (5%); therefore, for our data, the test rejected this hypothesis. The results of the t-test confirmed that the results of each experiment must be presented separately (Figure 6). At this moment, we have to rely only on 10 measurements of the strength produced in each experiment. The error bound up with that procedure will be further discussed. 

### 4.2. Regression Validity

One method allowing us to check the regression method validity by observing the R^2^ value. There exists a minimum value of R^2^ to accept the fit of the data to the Weibull distribution as calculated by the linear regression method [37]. When the sample size is equal to 20, the minimum R^2^ value is 0.894, and the R^2^ value must be over 0.855 for n = 10. In Figure 6, the value of R^2^ is presented, and the fit of lines found by regression were sufficient to accept all the presented data. For the elongation data, the fit was not sufficient to present the Weibull modulus found by regression.

### 4.3. Validity of Comparison between Distributions

Even considering the R^2^ test, the smaller sample size in the Weibull analysis resulted in an increased error [39]. To check if the results of the experiments (blue squares and purple diamonds in Figure 6 and Figure 7) can be compared with the reference data (red triangles in Figure 6 and Figure 7), we need to refer to the confidence intervals published earlier [34].

A comparison between two distributions (with known m and m′) of sizes N = 20 and N′ = 10 is possible with 95% confidence if [40]
1.889 < m′/m < 2.434.

For experiment 1, the m′/m = m_1_/m_0_ = 2.216. As this value is inside the confidence interval given above, there is 95% confidence in comparing the Weibull moduli of both of these distributions. For experiment 2, the m′/m = m_2_/m_0_ = 1.837, and thus we can compare it with the reference results with 90% confidence (where the confidence interval starts for 1.695 < m′/m as given at [40]). Thus, even considering the errors, we can confirm the quality improvement for both treatments with over 90% confidence.

### 4.4. Validation of Used Estimators

Fitting the correct distribution when the sample size is small should be done with the correct method. Several proposed estimators [38,40,42,43] and weighted linear regression [44,45] have been validated for Ex1 and Ex2. The results of the Anderson–Darling goodness-of-fit test are presented in Table 4.

The Anderson–Darling test rejects the dataset if the value *p* ≤ 0.05. It simply means that, with 95% confidence, the data do not follow the chosen distribution. Both tested datasets show a good fit, with the *p*-value being on the extraordinary level of 0.99 for Ex1 validating previous analysis. 

From Table 4, we can also conclude that the alteration of the estimators is not necessary, and the Anderson–Darling test did not show significant differences between all the tested methods.

### 4.5. Expected Error

To make further analysis easier, we rounded the m-values. For experiment 1 (see Table 4), the Weibull modulus was close to m_1_ = 68, and, for experiment 2, m_2_ = 56. The mean values of the elongation and UTS after treatment (Table 2 and Table 3) prove that—even accounting for the maximum error—both experiments improved the quality of the processed metal. 

One can query the exact expected error in the established value of the Weibull modulus. The goodness of fit results can indicate that the chosen distribution was fitted correctly, even if based on a small number of measurements. To be more precise in the error estimation, we considered the estimated value for Ex1. By the computer simulation, the standard deviation and confidence intervals were calculated and shown in published research [38,40,45]. The error found was about 33% when only 10 test-bars were used [38,45]. For confirmation of that value, the results of computer simulation with 50,000 cycles were used to estimate the bias and the error (the standard deviation of the mean Weibull modulus calculated by different methods). The results are presented in Table 5.

As we can see, the simulation results confirmed that the expected error was about 33% when 10 samples were used. By increasing the number of measurements to 20 (a common procedure), this error was decreased by 10%. The most unbiased method was number 2, used in the previous calculations for Ex1 and Ex2 (Figure 6 and Figure 7).

Let us consider the worst-case scenario for Experiment 1. If the m_1_ = 68 is overestimated by 33%, we will obtain a minimum value of the Weibull modulus close to 46. This value is still significantly higher than was observed for the reference samples, equal to 31—which is treated as given with sufficient certainty due to the higher number of data taken for calculations. m is often between 10 and 30 for gravity-filled castings [32], and the value m_0_ = 31 obtained without ultrasonication appears to be reliable and is not expected to be much higher.

Experiment 2 also showed improvement of the metal quality, which supports the hypothesis regarding the significant improvement in Ex1. In that case, if the error is maximal, the value will go down to about 40, which still can be seen as an improvement in the metal quality.

The mechanical properties, according to the Hall–Patch equation, are expected to improve as the grain size decreases. Using the prediction from Figure 2 and Figure 3, we can expect a better improvement of the metal quality in Ex1 than in Ex2. Those changes are reflected in the calculated Weibull modulus values—the highest value of 68 for Ex1 (with small scatter of data, the small error in Table 2) and a smaller value of 56 for Ex2. The gradual changes of m according to the grain size can be treated as another validation of the established m. Thus, the expected error is below the maximum possible value.

The Weibull modulus of UTS for good quality aerospace castings is expected to be between 50–100 [32]. The m values characterising the processed alloys presented here fulfil those conditions. 

### 4.6. Role of Degassing and Structure Changes in Metal Straightening

Two measurable changes can be discerned due to the sonication of the melt. The first one concerns the gas content, and the second concerns the grain size, reflecting changes on interfaces (forced wetting, undercooling, dispersion etc.). To find an answer as to which effect is dominant in the quality improvement obtained by the ultrasound, we need to recall the results of Ex5 (Table 2 and Table 3).

The dataset for Ex5 was produced following treatment by the contactless sonotrode in the setup, where the number of collapsing bubbles was not sufficient to cause grain refinement. In other words, most of the bubbles created oscillated in size but did not burst. Due to the lack of shock wave emissions, the expected de-agglomeration and particle dispersion could not happen, and thus we did not observe grain size reduction. Instead, processing by the contactless sonotrode caused degassing at the level of 0.06 cm^3^/100 g, without noticeable grain size change; this effect on its own did not improve the metal strength or elongation. Similar processing, with the same level of the gas removed during treatment followed by grain refinement, improved the metal properties significantly. 

Thus, the main mechanism involved in the metal strengthening observed in Ex1 and Ex2 was caused by a grain size decrease rather than decreased gas content. As a conclusion, it is necessary to achieve full cavitation at near-resonant frequency to cause changes at the interface that cause effective de-agglomeration and dispersion.

## 5. Conclusions

Contactless sonication provided significant changes in the observed microstructure of the cast alloy and decreased the grain size by up to 48%.The treatment was followed by a gas content decrease of up to 0.11 cm^3^/100 g.Ultrasonicated melts after casting exhibit improved the ductile properties, and the percentage of elongation increased maximally 1.7 times (from 16% to 28%).The processed alloy showed a great improvement in strength. The mean UTS was increased by about 17%. This was followed by changes in the calculated Weibull modulus from 31 to 68. Even considering the maximum possible error, the quality of the metal after treatment was significantly improved.

## Figures and Tables

**Figure 1 materials-14-04479-f001:**
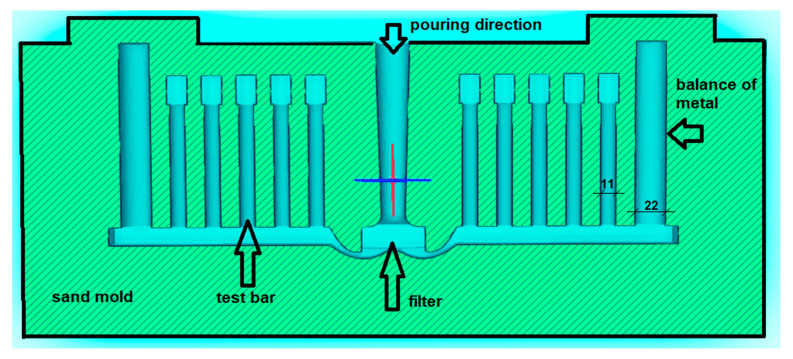
The draft of the sand mould used for casting of 10 test-bars for tensile strength tests. The two crucial diameters are given in mm, and the sketch scale is 1:1. Liquid metal is poured into the mould from the top opening; it passes the filter and feeds the 10 test-bars cavities (diameter 11 mm). The higher cavities on both ends (diameter 22 mm) balance the pressure to feed the mould correctly.

**Figure 2 materials-14-04479-f002:**
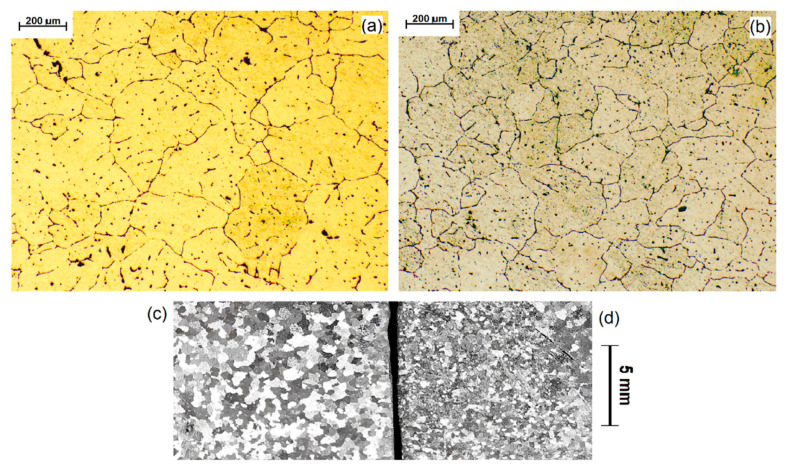
Experiment 1: (**a**) The grain size of untreated metal (208 ± 38 μm) and (**b**) after treatment (108 ± 19 μm) decreased by 48%, which is clearly indicated by a macro-scale photograph of a sample taken (**c**) before and (**d**) after ultrasonication.

**Figure 3 materials-14-04479-f003:**
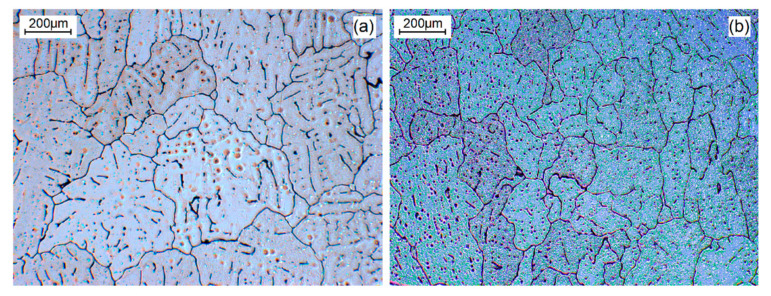
Experiment 2: (**a**) The grain size of untreated metal (204 ± 13 μm) and (**b**) after treatment (147 ± 4 μm) decreased by about 28%.

**Figure 4 materials-14-04479-f004:**
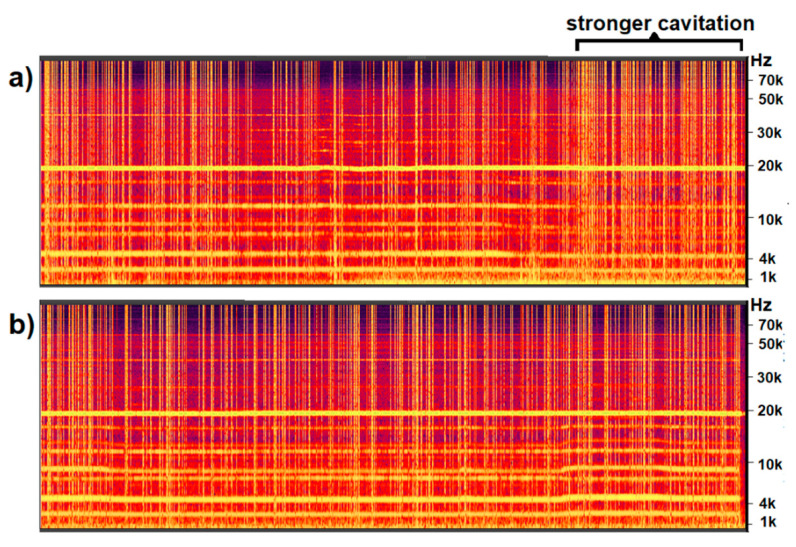
Spectrograms extracted from the recorded sound: (**a**) experiment 1, ultrasonication with frequency 18.86 kHz, and (**b**) experiment 2, ultrasonication with frequency 18.42 kHz. The time of recording in both cases was about 5 min long.

**Figure 5 materials-14-04479-f005:**
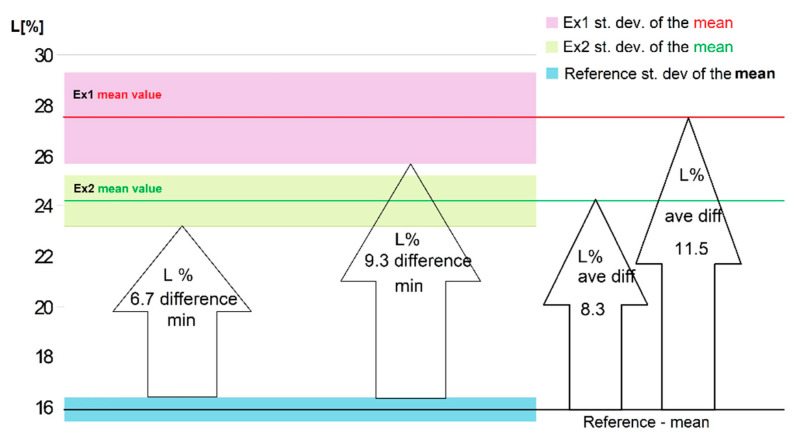
Measurements of elongation—the mean values and the error found in each experiment. The arrows show the minimum differences between elongation with and without treatment (difference min) or the difference between the mean values (average difference).

**Figure 6 materials-14-04479-f006:**
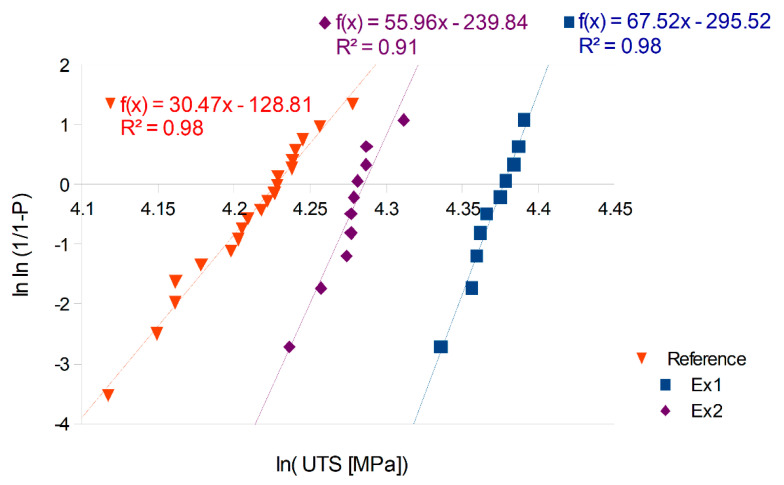
The Weibull modulus established by the linear regression.

**Figure 7 materials-14-04479-f007:**
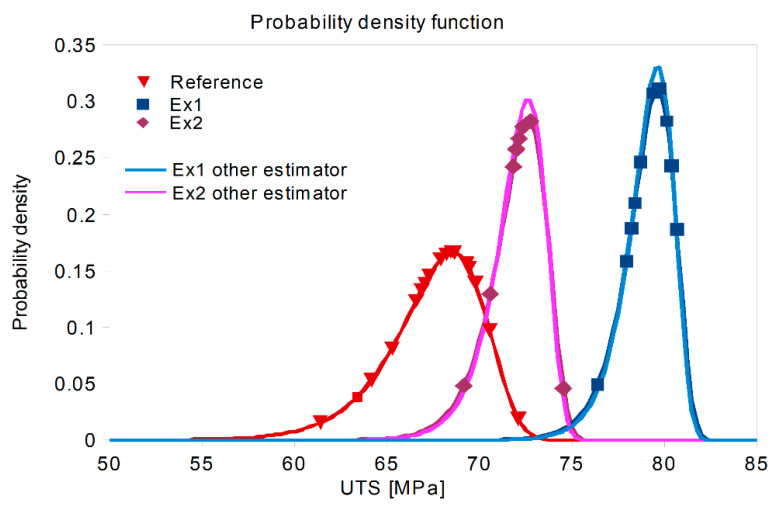
The probability density function (Weibull curve) of the reference data versus the results of both ultrasound treatments based on UTS measurements. The data points are overlayed on the calculated distribution indicated in Figure 6. Two lines labelled as “other estimator” show the other possible distributions, calculated with another estimator, as will be discussed further (method 3 in Table 4).

**Figure 8 materials-14-04479-f008:**
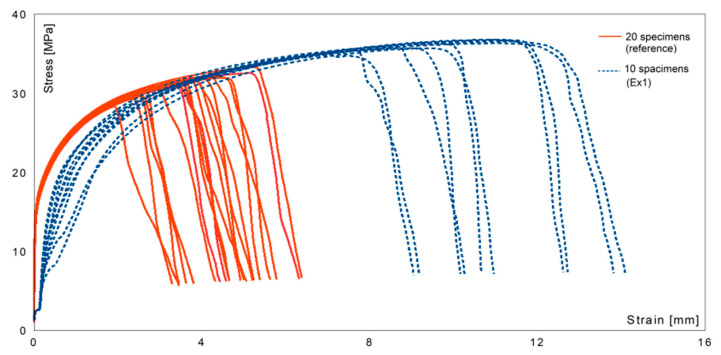
The stress versus the nominal strain curves for the specimens from the reference set and experimental (Ex1) set.

**Figure 9 materials-14-04479-f009:**
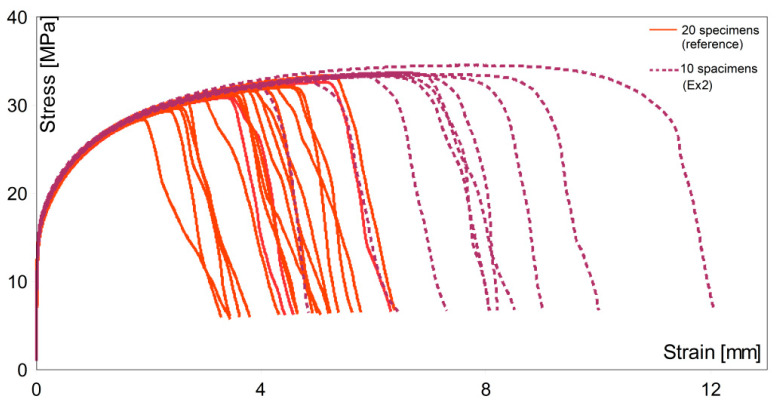
The stress versus the nominal strain curves for the specimens from the reference set and experimental (Ex2) set.

**Figure 10 materials-14-04479-f010:**
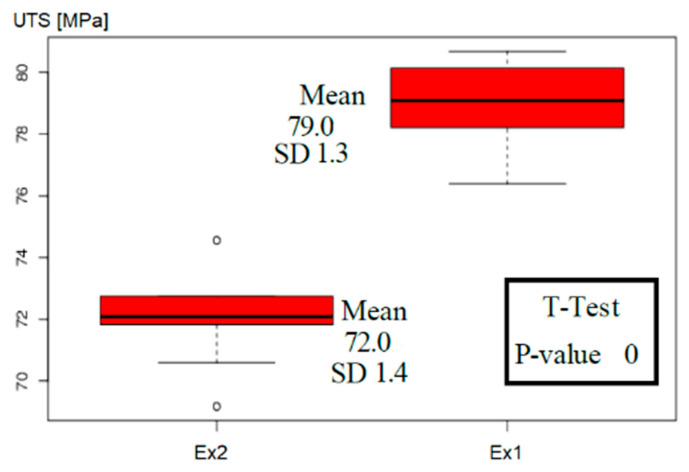
The results of two-sided t-tests of UTS measurements showing that Ex1 and Ex1 do not follow the same distribution. The test was done with 95% confidence.

**Table 1 materials-14-04479-t001:** Degassing achieved by contactless ultrasound treatment.

	Frequency (kHz)	Temperature (°C)	Gas Content before Treatment (cm^3^/100g)	Gas Content after Treatment (cm^3^/100g)	Gas Content Decrease (cm^3^/100g)	Gas Content Decrease (%)
Ex1	18.86	710	0.17	0.11	0.06	35%
Ex2	18.42	710	0.16	0.12	0.04	35%
Ex3	18.61	706	0.17	0.11	0.06	35%
Ex4	18.32	700	0.15	0.04	0.11	73%
Ex5 (*)	18.42	709	0.20	0.14	0.06	30%

(*) degassing only—no grain refinement produced.

**Table 2 materials-14-04479-t002:** Mean elongation before (L) and after (L′) treatment.

	L (%) Reference	L′ (%) after Treatment	L′/L	L′/L (*100%)
Ex1	15.9 ± 3.0	27.5 ± 6.5	1.73	173%
Ex2		24.2 ± 4.2	1.52	152%
Ex5 (*)		15.5 ± 6.5	0.98	98%

(*) degassing only, no grain refinement.

**Table 3 materials-14-04479-t003:** The mean UTS before (UTS) and after (UTS′) treatment.

	UTS (MPa) Reference	UTS′ (MPa) after Treatment	UTS’/UTS (*100%)
Ex1	67.4 ± 2.6	79 ± 1.3	117%
Ex2		72 ± 2.6	107%
Ex5 (*)		65.6 ± 4.5	98%

(*) degassing only, no grain refinement.

**Table 4 materials-14-04479-t004:** The Anderson–Darling (AD) goodness of fit test for m calculated by unweighted linear regression (LR) or weighted linear regression (WLR) with different estimators. Already presented values are marked by bold font.

No	Method	Used Estimator	Estimated m from UTS Data		AD Test Result (*p*-Value)		Approach Ref.
			Ex1	Ex 2	Ex1	Ex 2	
1	LR	P = (i − 0.5)/n	71.7	59.6	0.99	0.67	[42]
2	LR	P = (i − 0.348)/(n + 0.19)	**67.5**	**56.0**	0.99	0.64	[38]
3	LR	P = (i − 0.44)/(n + 0.12)	69.7	57.8	0.91	0.67	[43]
4	WLR	P = (i − 0.5)/n	59.6	56.3	≤0.05	0.66	[44,45]

**Table 5 materials-14-04479-t005:** The bias and the error comparison for methods (as in Table 4) used to estimate the Weibull modulus.

	Method	n	M	M′	Bias (M′/M)	Standard Deviation (M′)
Ex1	1	10	68	71.94	1.05	32.5%
	2			68.57	1.01	32.2%
	3			70.71	1.03	32.4%
Ex2	1	10	56	59.49	1.06	32.3%
	2			56.07	1.00	32.5%
	3			57.71	1.03	32.6%
Reference set	(*)	20	31	31.49	1.02	21.8%

(*) as in introduction for n = 20.

## Data Availability

Data is contained within the article.

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
