# Peer review of "Enhancement of Mechanical Properties of Pure Aluminium through Contactless Melt Sonicating Treatment"

_materials, 2021, doi:10.3390/ma14164479_

Round 1
Reviewer 1 Report
The article's content is focused on a vital issue, which is increasing casting quality by improvement of liquid metal quality. It is crucial for casting processes to keep that issue in mind. The article is well organized. The studies are interesting for readers, especially for casting company employees.
After analysis of content, some minor corrections are coming to mind that should be done.
- First of all, delete the part of template “0. How to use this template” from the manuscript.
- Consider expanding the introduction section by mentioning some other methods of casting quality improvement like gating system optimization or modification/ inoculation treatment of liquid alloy. There is a huge possibility to remove gas bubbles from the alloy only by the proper gating system design. Also, you mentioned using grain refiner in section 2, so it may be valuable to say a word about modification in the introduction part. Here are some proposed research works:
- Bruna, M.; Galčík, M. Casting quality improvement by gating system optimization. Arch. Foundry Eng. 2021, 1, 132–136. DOI: 10.24425/afe.2021.136089
- Dojka, R., Jezierski, J. & Tiedje, N.S. Geometric Form of Gating System Elements and Its Influence on the Initial Filling Phase. J. of Materi Eng and Perform 28, 3922–3928 (2019). https://doi.org/10.1007/s11665-019-03973-9
- Uludağ, M. ; Gurtaran, M. ; Dispinar, D. The Effect of Bifilm and Sr Modification on the Mechanical Properties of AlSi12Fe Alloy. Arch. Foundry Eng. 2020, 3, 99–104. DOI: 10.24425/afe.2021.136089
- Erzi, E.; Gürsoy, Ö.; Yüksel, Ç.; Colak, M.; Dispinar, D. Determination of Acceptable Quality Limit for Casting of A356 Aluminium Alloy: Supplier’s Quality Index (SQI). Metals 2019, 9, 957. https://doi.org/10.3390/met9090957
- Uludağ M., Gurtaran M., Dispinar D. (2019) Investigation of Casting Quality Change of A356 by Duration in Liquid State. In: Tiryakioğlu M., Griffiths W., Jolly M. (eds) Shape Casting. The Minerals, Metals & Materials Series. Springer, Cham. https://doi.org/10.1007/978-3-030-06034-3_16
- Uludağ, M. ; Gurtaran, M. ; Dispinar, D. The Effect of Bifilm and Sr Modification on the Mechanical Properties of AlSi12Fe Alloy. Arch. Foundry Eng. 2020, 3, 99–104. DOI: 10.24425/afe.2020.133337
- in line 110 change to “[20,29]”.
- in lines 107, 110, 132, 134, 135, 158… show in bracket the data about software and device according to the MDPI standards.
- in table 1 show also the percentage value of the difference in gas content.
- in lines 251-252 change to “(I)” and “(II)”.
- in line 267 it will be better to show the percentage value of the difference between the elongation of samples if the values are not at least twice as large.
- in line 292 like before, showing the percentage value is good enough.
- line 296 there is no reference to table 3 in the text. You should include it before the table after the headline.
- The discussion is valuable but in sections 4.3 and 4.4, some divagations are out of topic and unnecessary, for example, lines 390-398. Try to condense the contents of this part of the discussion and skip some mathematical considerations. According to the topic, the influence of the treatment with contactless sonotrode should be discussed, not statistical methods of analysis.
- line 489 the same as before, you should show only percentage values.
Reviewer 2 Report
This manuscript reports experimental results on microstructure and mechanical properties improvement through contactless sonication melt treatment of aluminium. The topic is of interest, and finding is sound, and the manuscript is well structured and presented. Some comments to improve the presentation are listed as the following:
Title: as mentioned in the manuscript, the treatment can improve both strength and ductility of the aluminium, not only strength. Therefore, suggest changing the title to “Enhancement of mechanical properties of pure aluminium through contactless melt sonicating treatment”
Abstract: abstract should focus on what the manuscript is about, and the key findings. The contents in lines 13-18 is the background knowledge of the ultrasonic melt treatment, not close related to the manuscript.
Introduction:
Line 44, “larger volumes” the comparative adjective is not properly used.
Lines 79-81, “From the Hall-Petch equation [24] it is expected that the reduced grain size will improve the metal quality. Also, degassing provided by the ultrasound should also improve the properties of alloys.” The statement is not precisely and accurately to explain the metal/alloy quality.
Lines 92-93, “The tensile strength (UTS) for gravity-filled castings is generally thought to be 10 to 30 while for aerospace castings it is expected to be between 50 and 100”. Authors should clarify the alloy for the UTS mentioned, and secondly provide the unit UTS.
Line 95, “the strength measurement and elongation tests” should come from tensile test, not separated?
Materials and Methods
Lines 100-101, it is suggested to provide internal diameter or thickness plus ED for crucible.
Lines 149-150, can authors explain why “The cavitation intensity decreases with the temperature increase and almost disappear at 720C”?
Results: this investigation is on the improvement of mechanical properties of the aluminium, therefore, add more experimental results such as add a couple of tensile curves, fractographies, etc can help to address it and highlight the finding.
